METHODS AND PROTOCOLS

# Use and Evaluation of a pES213-Derived Plasmid for the Constitutive Expression of *gfp* Protein in Pathogenic Vibrios: a Tagging Tool for *In Vitro* Studies

William A. Norfolk,[a] Erin K. Lipp[a]

[a]Department of Environmental Health Science, University of Georgia, Athens, Georgia, USA

**ABSTRACT** Insertion of green fluorescent protein (GFP) into bacterial cells for constitutive expression is a powerful tool for the localization of species of interest within complex mixtures. Here, we demonstrate and evaluate the efficacy of the pES213-derived donor plasmid pVSV102 (*gfp* Kn$^r$) as a conjugative tool for the tagging of *Vibrio* and related species (termed vibrios). Using a triparental mating assay assisted by the helper plasmid pEVS104 (*tra trb* Kn$^r$), we successfully tagged 12 species within the *Vibrionaceae* family representing 8 of the proposed clades. All transconjugant strains demonstrated bright fluorescence and were readily differentiable within complex mixtures of nontagged cells. Plasmid retention was assessed using persistence and subculture experimentation. Persistence experiments evaluated plasmid loss over time for nonsubcultured samples inoculated into antibiotic-free media and sterile artificial seawater, whereas subculture trials evaluated plasmid loss following one to four subculture passages. Strong plasmid retention (≥80%) was observed in persistence experiments for all transconjugant strains for up to 48 h in both antibiotic-free media and artificial seawater with the exception of *Vibrio cholerae*, which showed a substantial decline in media after 24 h. Subculturing experiments also demonstrated strong plasmid stability, with all transconjugant strains showing ≥80% retention after four subculture passages. The results of this research suggest that pVSV102 is a stable GFP plasmid for the tagging of a broad range of vibrios.

**IMPORTANCE** Prior research has suggested that the use of *Aliivibrio fischeri*-derived donor plasmids with the pES213 origin of replication may provide increased plasmid stability for the tagging of vibrios compared to *Escherichia coli*-derived p15A plasmids. Here, we present a structured protocol for conjugation-based tagging of vibrios using the pES213-derived plasmid pVSV102 and evaluate the plasmid stability of tagged strains. These methods and the resulting transconjugant strains provide important standardized tools to facilitate experimentation requiring the use of traceable vibrio strains. Furthermore, the determination of the species-specific plasmid stability provides an estimation of the anticipated level of plasmid loss under the given set of culture conditions. This estimation can be used to reduce the occurrence of experimental biases introduced by plasmid drift.

**KEYWORDS** green fluorescent protein, *Vibrio*, GFP tag

*Vibrio* spp. (or vibrios, a colloquial term used to describe all members of the family *Vibrionaceae*) are ubiquitous aquatic bacteria commonly found in marine, coastal, and estuarine habitats worldwide (1). As indigenous members of the aquatic community, vibrios exhibit a diverse range of preferential lifestyles, with individuals existing as free-living bacterioplankton, as constituents of biofilm communities, or in mutualistic or pathogenic associations with host organisms (2). Through these complex interactions, vibrios play an important role in the ecology of aquatic ecosystems through their

Address correspondence to Erin K. Lipp, elipp@uga.edu.

The authors declare no conflict of interest.

contributions to biogeochemical cycling, roles in the food web, beneficial symbioses, and as agents of disease (1, 2). Vibrios act as pathogens across a broad range of hosts, from economically important penaeid shrimps to critically endangered scleractinian corals and humans (1, 3–5). The ability to localize the physical association of vibrios in their environment or host in a controlled setting is an important tool for investigating their ecology and pathways of transmission.

The use of fluorescent molecules to label bacteria is common to visualize and localize cells or their expressed proteins in systems of interest (6–9). While numerous tagging molecules exist, green fluorescent protein (GFP) and its derivatives remain among the most popular due to their intrinsic stability and resistance to photobleaching (7, 10, 11). First isolated from the jellyfish *Aequorea victoria*, GFP is a protein that exhibits a bright green fluorescence when excited with blue/UV light (10–13). Since its discovery in 1962 (13) and adoption for use as a molecular marker in 1992 (14), GFP tagging methods have been optimized for various experimental outcomes ranging from localization of host-pathogen/vector interactions to detection of target gene expression (15–19).

Fluorescent tagging is a particularly useful approach when investigating the environmental or host-associated dynamics of indigenous microorganisms such as vibrios, where it is otherwise impractical to differentiate an introduced experimental strain from the existing population. GFP tagging allows for the visualization of specific strains within complex systems, which can elucidate potentially important inter- and intraspecies interactions with the microbial community, the environment, or within a host (6–8). When used in conjunction with specialized microscopy techniques and/or histopathology, GFP tags can provide vital spatial information on the colonization of pathogenic or symbiotic bacterial species within a host or movement among host tissues (16, 20, 21). Prior studies have successfully employed GFP-vibrios to model host-pathogen interactions in oysters (22, 23), lobsters (24), corals (25), fishes (16, 26), and *Caenorhabditis elegans* (as a model for human wound infection) (27), as well as host-symbiont interactions in the Hawaiian squid (*Euprymna scolopes*) (28, 29). While these studies demonstrate the utility of GFP tags for vibrio research, there is a need for standardized methods of tagging that can be applied to a range of different vibrios.

Prior research by Dunn et al. (6) and Sawabe et al. (7) successfully used conjugation-based methods to tag vibrio species. Dunn et al. (6) labeled *Aliivibrio fischeri* using a triparental mating assay with the helper plasmid pEVS104 (*tra trb* Kn$^r$) (30) and one of several pES213-derived (31, 32) donor plasmids. Shortly after, Sawabe et al. (7) employed a biparental mating assay using a single *Escherichia coli* strain carrying both the helper plasmid pEVS104 (*tra trb* Kn$^r$) and a p15A-derived (30) donor plasmid, pKV111 (*gfp* Cm$^r$) or pKV112 (*gfp* Cm$^r$ Er$^r$), to tag 39 different vibrios. While the work of Sawabe et al. (7) effectively demonstrated the broad efficacy of conjugation-based tagging methods in vibrios, Dunn et al. (6) noted decreased plasmid stability when using *E. coli*-based p15A donors compared to *A. fischeri*-based pES213 donors. This finding suggests that pES213-derived donor plasmids may improve the retention of GFP tags in vibrios. Several subsequent studies have successfully employed pES213-derived donor plasmids for the creation of stable GFP tags in *A. fischeri* (33) and the *Vibrio* spp. *V. harveyi* (18, 34), *V. parahaemolyticus* (35, 36), *V. coralliilyticus* (37), *V. aestuarianus* (22), and *V. tapetis* (38). However, a formal side-by-side comparison of conjugation methods and the acquired plasmid retention is needed to standardize these methods across different vibrios.

Through this research, we present a simple protocol for GFP tagging of vibrios using a pES213-derived donor plasmid system and evaluate the plasmid retention of transconjugant strains. Through the combined use of the helper plasmid pEVS104 (*tra trb* Kn$^r$) (30) and the pES213-derived donor plasmid pVSV102 (*gfp* Kn$^r$) (6), we successfully tagged species across a range of the known *Vibrionaceae* clades (Tables 1 and 2). The efficacy of the GFP tags was evaluated with subsequent culture-based methods and fluorescence microscopy to determine the experimental limitations of the species-specific GFP retention.

**TABLE 1** Kanamycin lethal limits, GFP transfer concentrations, preferred culture medium type, and GFP conjugation outcomes and for all tested vibrios

| Species | Strain | MIC ($\mu$g mL$^{-1}$)[a] | Stress concn ($\mu$g mL$^{-1}$)[b] | CFU added to mating mix | Preferred culture medium[c] | Conjugation outcome | Strain isolation source | Strain reference |
|---|---|---|---|---|---|---|---|---|
| Photobacterium damselae | ATCC 33539 | 25 | 15 | $3.8 \times 10^7$ | LBS 3% | + | Damselfish skin ulcers, USA | 52 |
| Vibrio alginolyticus | ATCC 17749 | 100 | 75 | $4.6 \times 10^7$ | TCBS | + | Spoiled horse mackerel, Japan | 53 |
| Vibrio anguillarum | ATCC 19264 | 50 | 35 | $3.0 \times 10^7$ | LBS 3% | + | Ulcerous lesion in cod | 54 |
| Vibrio campbellii | ATCC 25920 | 50 | 35 | $6.3 \times 10^7$ | LBS 3% | + | Seawater | 55 |
| Vibrio coralliilyticus | ATCC BAA-450 | 50 | 35 | $3.3 \times 10^4$ | LBS 3% | + | Infected coral, Zanzibar | 4 |
| Vibrio cholerae | ATCC 14035 | 100 | 75 | $2.9 \times 10^6$ | LBS 3% | + | Enteric illness in humans | 56 |
| Vibrio furnissii | ATCC 35016 | 25 | 15 | $4.2 \times 10^6$ | LBS 3% | − | Human feces, Japan | 57 |
| Vibrio harveyi | ATCC 14126 | 35 | 25 | $7.1 \times 10^6$ | TCBS | + | Deceased luminescent amphipod, USA | 58 |
| Vibrio mediterranei | ATCC 43341 | 125 | 100 | $2.3 \times 10^7$ | LBS 3% | + | Sediment, Spain | 59 |
| Vibrio parahaemolyticus | ATCC 43996 | 50 | 35 | $3.7 \times 10^7$ | LBS 3% | + | Cockles (marine bivalve) | 60 |
| Vibrio pelagius | ATCC 25916 | 35 | 25 | $4.4 \times 10^5$ | TCBS | + | Seawater | 55 |
| Vibrio splendidus | ATCC 33869 | 75 | 50 | $1.16 \times 10^7$ | LBS 3% | + | Seawater, USA | 61 |
| Vibrio tubiashii | ATCC 19109 | 25 | 15 | $3.1 \times 10^6$ | LBS 3% | − | Juvenile hard clams | 62 |
| Vibrio vulnificus | ATCC 27562 | 50 | 35 | $6.2 \times 10^7$ | LBS 3% | + | Human blood, USA | 63 |

[a]The MIC concentration is defined as the minimum concentration of kanamycin that produced total growth inhibition of the target Vibrio.
[b]The stress concentration is defined as the concentration of kanamycin that creates a stressful but nonlethal environment for the growth of the target Vibrio. This concentration was utilized to facilitate transfer for the GFP plasmid into the target species.
[c]The preferred culture medium produces noninhibited growth under standard culture conditions and reduces the occurrence of bacterial swarming.

**TABLE 2** Description of the plasmids and carrier strains utilized in this study

| Plasmid designation | Carrier bacterium | Usage | Genes carried | Antibiotic resisted | CFU added to mating mix | Plasmid reference |
|---|---|---|---|---|---|---|
| pEVS104 | *E. coli* | Helper | *tra, trb* | Kanamycin (Kn$^r$) | $6.0 \times 10^6$ | 30 |
| pVSV102 | *E. coli* | Donor | *gfp* | Kanamycin (Kn$^r$) | $6.3 \times 10^6$ | 6 |

## RESULTS

**GFP expression.** This method successfully transferred GFP tags to 12 out of 14 tested vibrios (Table 1). All successfully tagged species showed strong conjugation efficiency (see Table S1 in the supplemental material) when mated triparentally on kanamycin-amended media equivalent to the stress concentration designated in Table 1. Tagged species consisted of important human and animal pathogens and represented 8 of the 23 proposed clades of the *Vibrionaceae* family (39). Bright GFP expression was observed in all transconjugants, allowing them to be readily differentiated from nontagged background vibrios within complex mixtures (Fig. 1, Fig. S14 and S15). No evidence of interspecies self-mobilization of pVSV102 was observed in the absence of antibiotic stress (Table S2). GFP expression was retained in all tagged strains following revival from −80°C frozen stocks.

**Persistence of GFP retention.** Plasmid retention of all transconjugant strains was assessed over time in media and artificial seawater to estimate the level of plasmid loss that occurred under growth and stagnation conditions, respectively. In antibiotic-free media, mean GFP retention for all transconjugants was ≥80% after 48 h, excluding *V. cholerae*, which showed a mean retention of only 36.8%. After 5 days, retention varied by species, with *Photobacterium damselae*, *V. alginolyticus*, *V. anguillarum*, *V. campbellii*, *V. parahaemolyticus*, and *V. vulnificus* showing minor plasmid loss with a mean GFP retention of 90 to 99%. Moderate loss was observed in cultures of *V. splendidus*, *V. mediterranei*, and *V. pelagius*, with a mean retention of 80 to 90%. Substantial loss was observed in

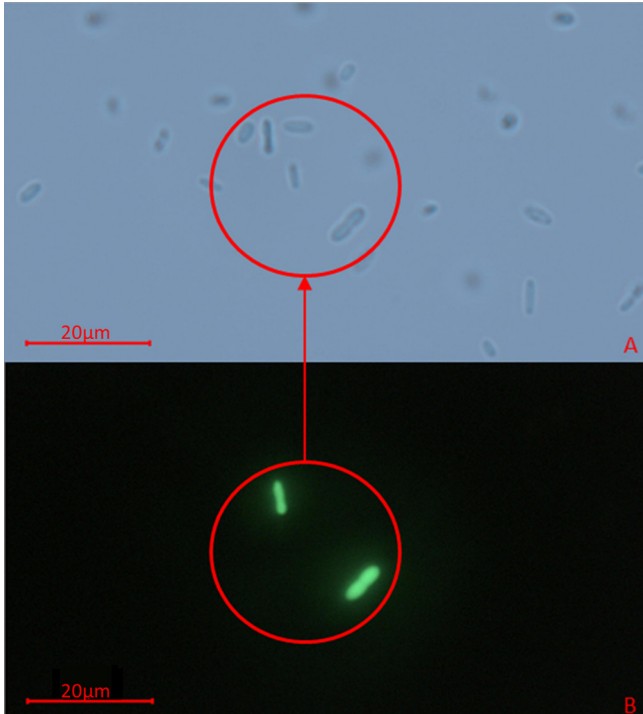

**FIG 1** Differentiation of GFP-tagged *V. alginolyticus* within a complex mixture of vibrios. The mixture contains equal parts *V. alginolyticus* (GFP), *V. campbellii*, *V. parahaemolyticus*, *V. harveyi*, and *V. vulnificus*. Panels A and B compare the same micrograph under light microscopy and fluorescence microscopy (495 nm excitation wavelength), respectively, at ×1,000 magnification.

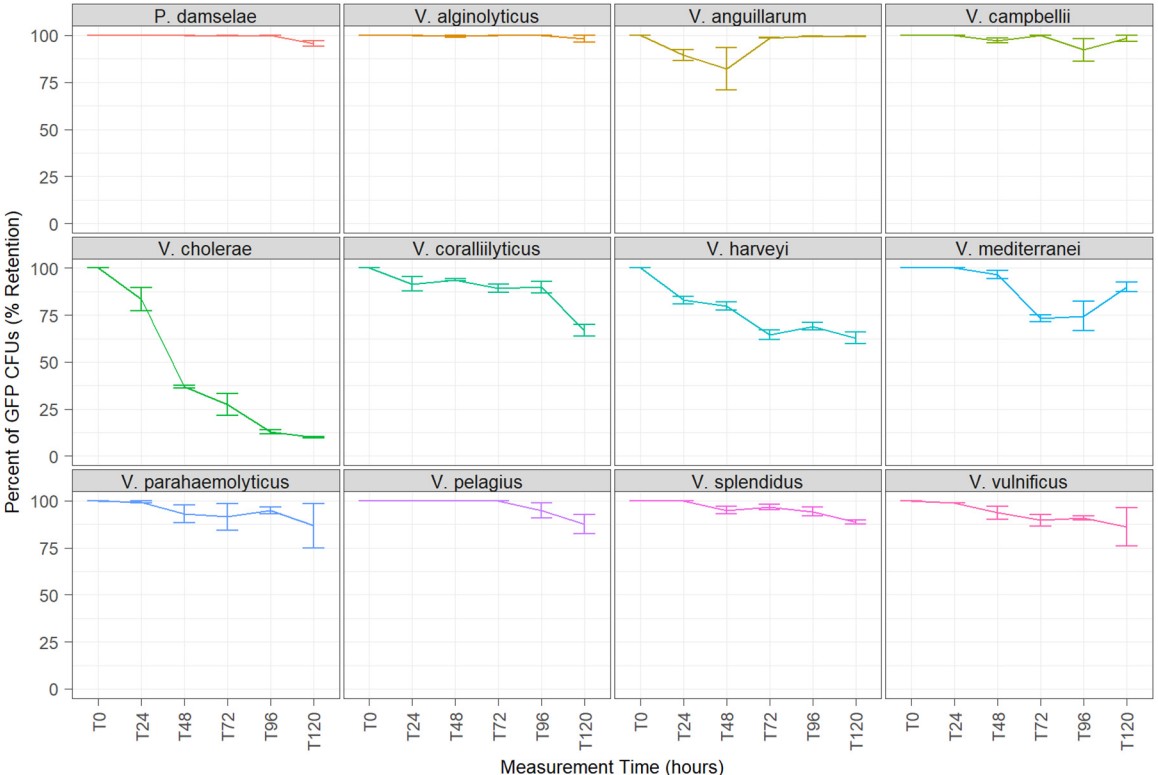

**FIG 2** Persistence evaluation of GFP retention in antibiotic-free media (LBS 3%) for 5 days of observation at 28°C. Values indicate the percentage of fluorescent CFU (y axis) observed at each time point (x axis). Error bars indicate the sample range (*N* = 2).

cultures of *V. cholerae*, *V. coralliilyticus*, and *V. harveyi* at <70% retention (Fig. 2). In the presence of antibiotics (300 μg mL⁻¹ kanamycin), which maintained selective pressure, 100% GFP retention was observed after 5 days of growth for all tested species, excluding *V. mediterranei*, in which retention declined to 90.7% by day 3 (Table S3).

In artificial seawater (ASW), all tagged vibrios showed a mean retention of ≥80% after 48 h. Moderate GFP loss (80 to 90% retention) was observed in *V. cholerae* and *V. vulnificus*, minor loss (90 to 99% retention) was observed in *P. damselae*, *V. coralliilyticus*, *V. harveyi*, and *V. mediterranei*, and no loss (100% retention) was observed for *V. anguillarum*, *V. alginolyticus*, *V. campbellii*, and *V. parahaemolyticus* (Fig. 3). *V. pelagius* and *V. splendidus* were not recoverable in ASW beyond 24 and 48 h, respectively.

**GFP retention during subculture.** Subculture experiments assessed the retention of GFP plasmids following multiple passages in antibiotic-free media to determine the effect(s) of culture regrowth on plasmid loss. GFP was retained at ≥80% in transconjugant strains for the duration of the experiment (up to four passages). After four passages, *V. alginolyticus* and *P. damselae* showed moderate plasmid loss (80 to 90% retention). Minor loss (90 to 99% retention) was observed in cultures of *V. anguillarum*, *V. campbellii*, *V. cholerae*, *V. harveyi*, *V. mediterranei*, *V. pelagius*, and *V. vulnificus*. No loss (100% retention) was observed in *V. parahaemolyticus*, *V. coralliilyticus*, and *V. splendidus* (Fig. 4). After one passage, all strains demonstrated 100% retention, except for *P. damselae*, which began at 92% retention (Fig. 4).

## DISCUSSION

Vibrios play multifaceted roles within marine and coastal ecosystems, through symbioses, interspecies competition, and pathogenicity (1, 2). The ability to localize vibrios of interest on or within host tissues is critical for helping to understand the ecological mechanisms that can influence these relationships. Of keen interest for this work was

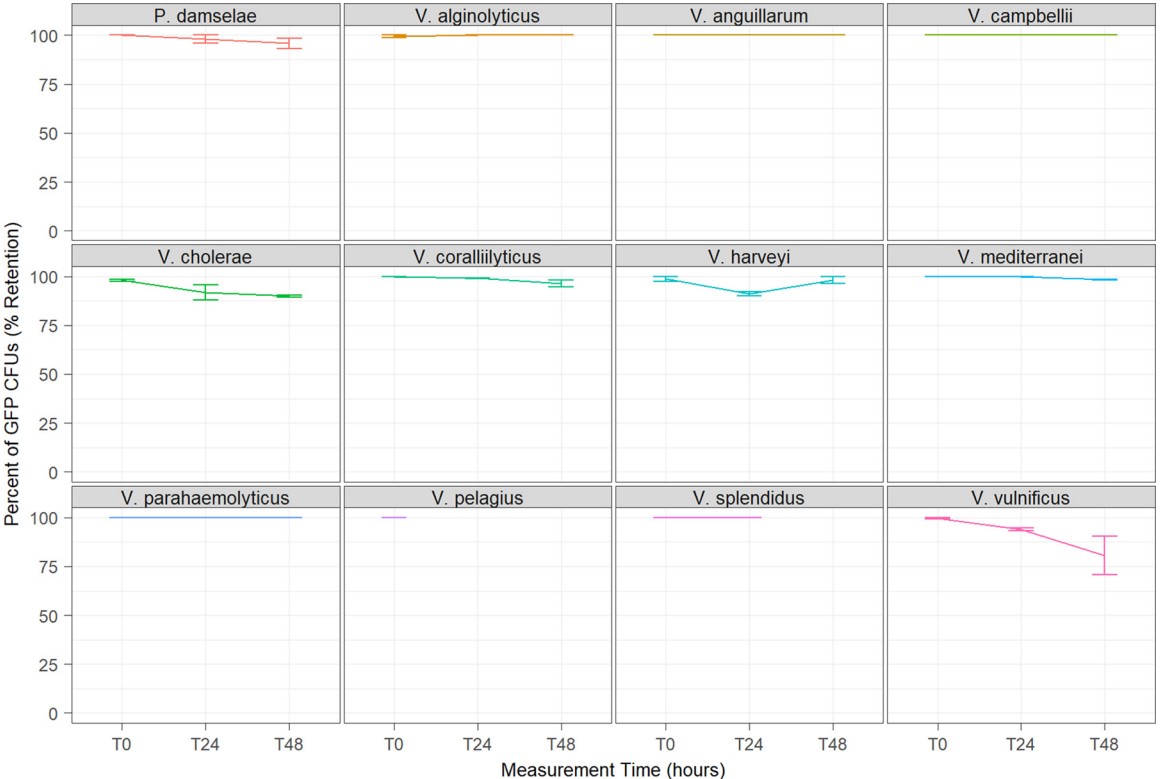

**FIG 3** Persistence evaluation of GFP retention in antibiotic-free seawater (sterile ASW) for 2 days of observation at 28°C. Values indicate the percentage of fluorescent CFU ($y$ axis) observed at each time point ($x$ axis). Error bars indicate the sample range ($N = 2$). Cultures of *V. pelagius* and *V. splendidus* were not recoverable in ASW beyond $T_0$ and $T_{24}$, respectively.

to optimize a GFP-tagging system across a wide range of vibrios and assess its efficacy for use in experiments designed to track host colonization of vibrios from ambient seawater.

We used a pES213-derived donor plasmid system to revisit several important vibrios previously tagged by Sawabe et al. (7) using a p15A-derived system. Prior research indicates that *A. fischeri*-based plasmids containing the pES213 origin of replication may produce more stable GFP expression in vibrios than *E. coli*-based p15A-derived plasmids (6). Using this system, successful transconjugant vibrios were created from 12 of the 14 tested species, covering 8 clades (39). These species include several understudied vibrios such as the coral pathogens *V. mediterranei* and *V. coralliilyticus*, the wide-host-range pathogens *V. alginolyticus* and *V. harveyi*, and the better-studied human pathogens *V. vulnificus* and *V. parahaemolyticus*, demonstrating a broad application of this system within the *Vibrionaceae* (4, 5, 18, 40). Furthermore, we expand on the Sawabe et al. (7) list of tagged vibrios through our addition of the fish pathogen *V. anguillarum* and the well-known human pathogen *V. cholerae* (5, 16). It should be noted that while this research prioritizes the use of pVSV102 in contrast to p15A-derivatives, other plasmid alternatives such as pRL1383a (41, 42), pUTat (43), pET28a (44), and pRK600 (25) have also been shown to persist stably in vibrios.

Evaluation of the transconjugant strains showed that the pES213-derived donor plasmid pVSV102 conferred bright GFP fluorescence when conjugated triparentally with the helper plasmid pEVS104. This fluorescence allowed all tagged vibrios to be readily differentiated and localized within complex bacterial mixtures. No evidence of interspecies self-mobilization was observed in mixtures of GFP *V. parahaemolyticus* and non-GFP *V. cholerae* and *V. vulnificus*, suggesting that pVSV102 has a low likelihood of interspecies mobilization in the absence of antibiotic stress (Table S2). Despite this observation, it should be noted that pVSV102 has been found to mobilize in conjunction

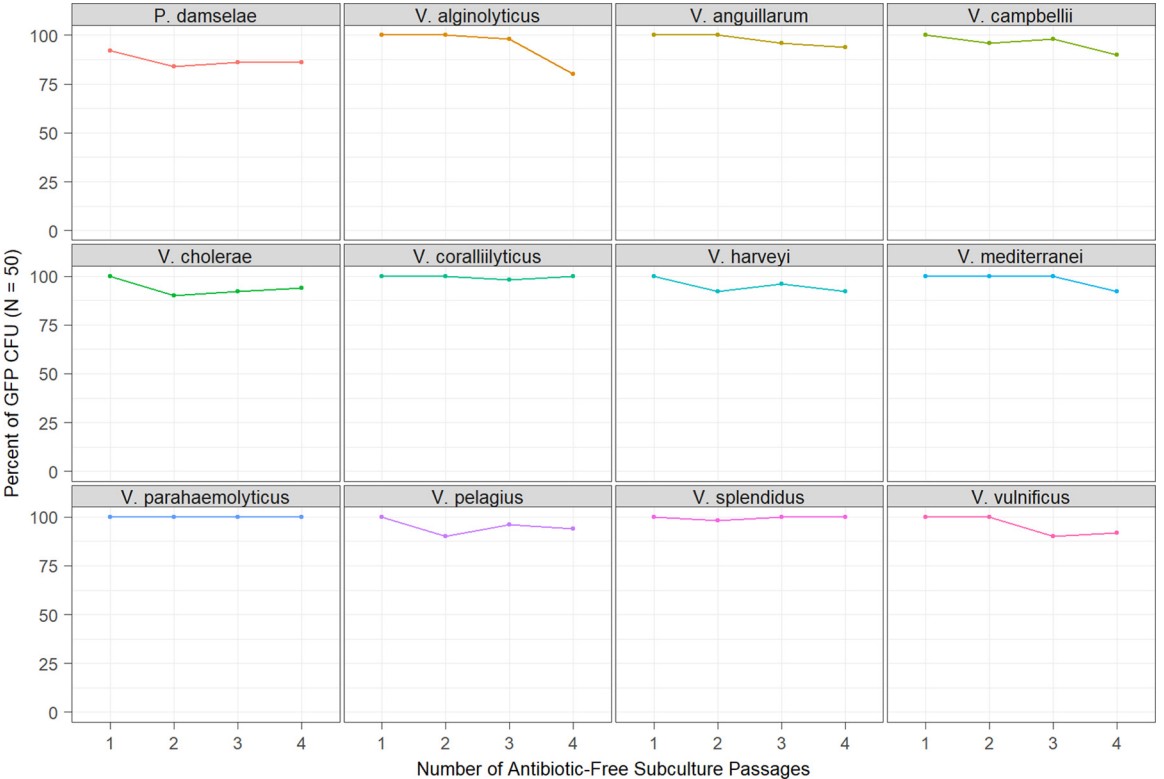

**FIG 4** Subculture evaluation of GFP retention following sequential passages in antibiotic-free media. Values indicate the percentage of patched CFU ($N = 50$) that successfully grew on 300 $\mu$g mL$^{-1}$ kanamycin-amended agar (y axis) following each level of subculture passages (x axis).

with the self-mobilizable *A. fischeri* plasmid pES100 (32), suggesting that vibrio strains carrying this plasmid or homologous plasmids may enable nontarget plasmid transfer in the absence of antibiotic stress.

Plasmid retention in pVSV102 provided equivalent or greater GFP stability in vibrios compared to p15A-derived plasmids (7). At least 80% GFP retention was observed in all target recipients, excluding *V. cholerae*, following 48 h of growth in antibiotic-free media. A gradual increase in plasmid loss was observed between days 3 and 5 in several species, namely, *V. coralliilyticus*, *V. harveyi*, and *V. mediterranei*. This loss was consistent with previously reported observations from Sawabe et al. (7), who noted a similar reduction by day 4 of experimentation. Despite this loss, these GFP strains remained >60% retentive throughout the entirety of the experiment, compared to 30 to 40% retention observed by Sawabe et al. (7), suggesting increased stability with a pES213-derived donor. Furthermore, several strains in the present work (*P. damselae*, *V. alginolyticus*, and *V. campbellii*) demonstrated near-complete retention (≥95%) after 5 days of growth, suggesting that high plasmid stability may exceed this duration depending on the target species of interest. It should be noted that the GFP retention reported by Sawabe et al. (7) was determined using direct microscopy counts, whereas this study utilized culture-based detection. Thus, these two values may not be directly comparable in all instances.

Under the nutrient-limiting conditions of ASW, persistence of GFP retention was ≥95% after 48 h in 10 of the 12 species, excluding *V. cholerae* and *V. vulnificus*, which retained 90% and 81%, respectively. Increased plasmid stability in ASW compared to media is likely related to decreased cellular growth under these conditions, which has also been shown for lower incubation temperatures (45). Based on these patterns, we hypothesize that ASW-maintained cultures would show a minor but progressive decrease in plasmid stability if allowed to persist beyond 48 h but would remain more retentive than media-maintained cultures. Interestingly, *V. vulnificus* demonstrated

similar patterns of GFP loss under ASW persistence as observed in antibiotic-free media growth, suggesting this species may be less amenable to ASW experimentation. This discrepancy may be due to stress induced by the lower salinity tolerance of *V. vulnificus* compared to other vibrios (46).

The results of subculture experimentation were consistent with those observed in persistence trials with, ≥80% retention of the GFP plasmid observed in all transconjugants following four subculture passages (the maximum tested). These results suggest that pES213-derived plasmids are amenable to experimental methods requiring subculturing to prepare samples. Though it should be noted that while pVSV102 was resistant to loss during subculturing, some strains showed a small decline in plasmid retention, especially between passages 3 and 4 (Fig. 4), suggesting that experimental use of such strains should limit the number of subcultures to reduce the risk of retention bias.

Under the scope of the present research, the mechanism(s) contributing to *V. cholerae* plasmid loss in persistence experiments are unclear. As the only nonhalophilic bacterium tested, we hypothesize that this loss may be related to metabolic stress from the medium, which is known to reduce plasmid stability (47, 48). Furthermore, it has been shown that some strains of *V. cholerae* possess plasmid defense mechanisms which may further contribute to the loss of GFP tags in the absence of antibiotic stress (49). Optimal stability may be achievable in *V. cholerae* through modification of the methods by reducing the salt content of the mating, persistence, and purification media used to facilitate transfer; however, further experimentation is required. Of the tested strains, *V. furnissii* and *V. tubiashii* were the only species that were unsuccessful in acquiring the GFP plasmid. These two species along with *P. damselae* were noted to have the lowest kanamycin tolerance threshold of all tested vibrios. During experimentation, nontolerant species (MIC of kanamycin of ≤50 $\mu$g mL$^{-1}$) showed increased sensitivity to the stress concentration used to transfer the GFP. While tagging was achievable in nontolerant species (as evidenced by *P. damselae*), a more highly resolved determination of the kanamycin MIC for the stress concentration may be required for these or similar tolerance-level vibrios.

Based on the observed patterns of plasmid stability, experimentation using these strains should be limited to exposure durations where the species-specific retention is ≥80%. Keeping within this range would ensure that major stability biases are avoided. To apply this work to experiments where quantification is required, assays to specifically determine the expected plasmid loss for the target species under the defined culture/experimental conditions (e.g., media, incubation temperature, oxygenation level, and number of anticipated subculture passages) are needed to account for plasmid drift over the course of the experiment. For presence-absence experimentation, microscopic visualization of the tagged strains is possible even at low levels of GFP retention (*V. cholerae* remained detectable microscopically at ~10% retention), allowing for potentially longer-duration experiments. However, differentiation of GFP-tagged bacteria may be difficult to discern at these levels if the matrix of interest is highly complex, such as within host tissues or attached to a substrate.

While the use of GFP for bacterial localization can provide important insights into the ecology of a species, GFP-tagged bacteria are not true wild-type strains. The creation of a transconjugant bacterium intrinsically changes the biology of the individual. This alteration can give rise to physiological, behavioral, and/or morphological differences in phenotype that may not be representative of wild-type strains. Such differences have been observed previously (50) and were noted in our media retention experiments, where the colony size of *V. cholerae* increased when the GFP plasmid was lost compared to colonies that were retentive, suggesting increased fitness for nonretentive cells (Fig. S16). While some differences are not unexpected, care is needed when designing experiments using GFP-tagged strains to ensure that any conjugation-induced experimental biases are accounted for prior to the start of the research.

**Conclusion.** The results of this research demonstrate the utility and stability of pVSV102 as a conjugative tool for the GFP tagging of vibrios. The methods present a standardized protocol for conjugation-based transfer of pVSV102 using a tri-parental

mating assay with the helper plasmid pEVS104 and kanamycin-amended media. Using these GFP strains, researchers can better design experiments to identify and/or describe potential vector species, reservoirs, bacterial aggregation patterns, and chemotaxis, which can be used to better understand the ecology and/or manage the pathogenic burden of vibrios.

## MATERIALS AND METHODS

**Strain selection.** Experimental vibrio strains were obtained from the American Type Culture Collection (ATCC) (Table 1). *E. coli* strains carrying the helper and donor plasmids were created and graciously provided by the Stabb Lab (Eric Stabb, University of Illinois Chicago, Chicago, IL) (Table 2). Experimental vibrios were selected to represent a range of phylogenetic clades, with particular emphasis given to type strains, where possible. All storage cultures were maintained at −80°C in 20% glycerol (vol/vol, final concentration) prior to the start of the experiments.

**Kanamycin tolerance assessment.** The helper and donor plasmids selected in this study carry kanamycin (Kn$^r$) resistance (6, 30). Thus, the strain-specific tolerance of all experimental vibrios was assessed to determine a MIC of this antibiotic. Assessment of the MIC was used to establish a stress concentration, or the concentration at which the antibiotic becomes detrimental but does not completely inhibit growth (Table 1). Overnight cultures of the frozen vibrio stocks were grown in antibiotic-free lysogeny broth (LB; Sigma-Aldrich, Miller formulation) amended with 3% wt/vol NaCl (here termed LBS 3%) at 30°C with 100 rpm shaking agitation (New Brunswick Scientific, C24 incubator shaker). Cultures were then streaked for isolation onto thiosulfate citrate bile salts sucrose agar (TCBS) or LBS 3% agar, each amended with 50 $\mu$g mL$^{-1}$ kanamycin. The use of TCBS or LBS 3% agar was determined by species-specific preference to each medium as noted in Table 1. Any strains that successfully grew at 50 $\mu$g mL$^{-1}$ were deemed Kan tolerant and subsequently cultured at 75, 100, 125, 150, and 200 $\mu$g mL$^{-1}$ concentrations of kanamycin to establish a lethal limit. Any strains that did not successfully grow at 50 $\mu$g mL$^{-1}$ were considered nontolerant and were subsequently cultured at 2, 5, 10, 25, and 35 $\mu$g mL$^{-1}$ to determine a lower tolerance threshold. The stress concentration for each species was calculated as a midpoint of the highest antibiotic concentration at which strains would grow in culture and the lowest lethal concentration (Table 1). The stress concentration was later utilized in the mating assay as the baseline antibiotic concentration for the transfer of the GFP plasmid.

**GFP conjugation culture preparation.** To begin the mating assay, all bacterial stocks were cultured overnight (18 h) in broth to ensure adequate growth of the strain and/or retention of the plasmid. Vibrio stocks were cultured in 3 mL of antibiotic-free LBS 3% at 30°C with 100-rpm shaking agitation. *E. coli* stocks were cultured in 3 mL LB broth (Sigma-Aldrich, Miller formulation) amended with 40 $\mu$g mL$^{-1}$ kanamycin at 35°C with 100-rpm shaking agitation. Following incubation, 1 mL of vibrio and *E. coli* cultures were pelleted by centrifugation at ~4,000 × $g$ for 2 min and then resuspended in 1 mL of sterile 1× phosphate-buffered saline (PBS). This procedure was repeated twice to wash cells and remove residual media. Then, 100 $\mu$L (CFU reported in Tables 1 and 2) of the washed helper, donor, and target recipient was removed and combined in a 1.5-mL microcentrifuge tube. The combined aliquots were centrifuged at ~4,000 × $g$ for 2 min to pellet the cells, and the supernatant was discarded. The pelleted cells were resuspended in 10 $\mu$L of fresh antibiotic-free LB broth amended with 2% wt/vol NaCl (LBS 2%). This salt concentration allowed growth of both vibrio and *E. coli*. The reduced volume of this final suspension was selected to increase the concentration of cells.

**Tri-parental mating assay.** Following culture preparation, the resuspended mixed bacterial pellet was spot-plated onto a thick (~10 mm) LBS 2% plate amended with kanamycin, equivalent to the stress concentration determined in the kanamycin tolerance evaluation for each vibrio strain (Table 1). The mating mixtures were incubated at 28°C for 24 to 72 h and checked daily with a 495-nm blacklight for fluorescence. Extended-duration incubation was utilized to account for the reduced rate of growth of the target vibrio under the given level of kanamycin stress. Mating mixtures that successfully produced green fluorescent patches within the cell masses were indicative of successful transfer of the GFP plasmid (Fig. S1). Mixtures were then streaked onto TCBS plates amended with 300 $\mu$g mL$^{-1}$ kanamycin to remove the helper and donor *E. coli* strains, which do not grow well on this medium, and ensure that the plasmid was retained within the vibrio culture. *Vibrio* species deemed nontolerant during the kanamycin tolerance experiment were sequentially streaked first onto 100, then 200, and finally 300 $\mu$g mL$^{-1}$ kanamycin TCBS plates to ensure that the antibiotic stress did not overwhelm the target *Vibrio*. Successful removal of *E. coli* strains was confirmed using tandem growth on modified mTEC agar (Difco, Fischer Scientific), an *E. coli*-specific medium. GFP transfer was confirmed for all purified vibrio strains using fluorescence microscopy (Olympus BX41 fluorescence microscope) (Fig. S2 to S13). Conjugation efficacy was assessed for 24 mixtures of each species of target vibrio (Table S1). Successfully tagged strains were cultured in LBS 3% broth amended with 300 $\mu$g mL$^{-1}$ kanamycin overnight at 30°C. Broth cultures were stored long term at −80°C in a 1:1 mixture of 40% glycerol (20% final concentration) and the kanamycin amended LBS 3% broth. It should be noted that while TCBS agar is valid for the removal of *E. coli*, this medium does not always produce optimal growth for some vibrio species; thus, working stocks of these cultures should be maintained on LBS amended with 300 $\mu$g mL$^{-1}$ kanamycin once isolated. For vibrios that are not amenable to growth on TCBS, prior research has successfully utilized auxotrophic *E. coli* strains to enable selective removal following conjugation (51).

**GFP persistence.** To determine the persistence of GFP plasmids in nonsubcultured samples, all transconjugant strains were grown in the absence of antibiotic stress, and the level of plasmid loss was

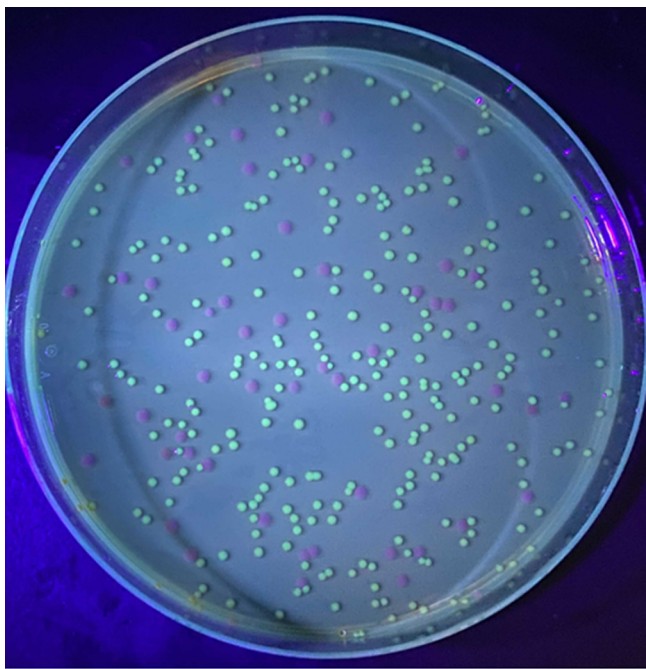

**FIG 5** Assessment of *V. cholerae* GFP retention in persistence experimentation. Bright green coloration indicates retentive CFU, whereas purple coloration indicates loss of GFP. Image taken at 24 h of growth with the aid of a 495-nm blacklight.

measured over time. Plasmid loss was evaluated as the percent loss of fluorescent CFU following growth in a long-term broth culture and persistence in sterile ASW. Long-term broth cultures were maintained in 4 mL of antibiotic-free LBS 3% for 5 days (120 h). ASW cultures were maintained in 10 mL of 0.2-$\mu$m filter-sterilized Instant Ocean (35 practical salinity units [PSU]) for 2 days (48 h). The larger volume was selected for ASW trials to stabilize the cultures under low-nutrient conditions, which were observed to have low survivability at the 4-mL volume (used in medium experiments). All cultures were revived from $-80°C$ storage and incubated overnight in LBS 3% amended with 300 $\mu$g mL$^{-1}$ kanamycin at 30°C with 100 rpm of shaking agitation. Fluorescence was confirmed for all parent cultures using fluorescence microscopy. Parent cultures were pelleted by centrifugation at $\sim$4,000 $\times$ $g$ for 2 min and then resuspended in sterile 1$\times$ PBS, in duplicate, to remove any excess kanamycin before the start of the experiment. Then, 100 $\mu$L of washed cells was inoculated in 4 mL ($\sim$2.5 $\times$ 10$^6$ CFU/mL) of antibiotic-free LBS 3% and 10 mL ($\sim$10 $\times$ 10$^6$ CFU/mL) of sterile ASW. Cultures were maintained at 28°C under 100 rpm of shaking agitation for up to 5 days. Daily, a 100-$\mu$L aliquot of each culture was removed from the incubator, serially diluted (10-fold), spread-plated with glass rattler beads (Zymo Rattler plating beads, 4.5 mm) onto agar plates (species-specific medium preference; see Table 1), and incubated at 30°C overnight, in duplicate. Plates were examined the following day ($\sim$18 h), and the number of fluorescent and nonfluorescent CFU were counted with the aid of a 495-nm blacklight (Fig. 5). GFP loss was calculated as the percent reduction of fluorescent CFU over time.

**GFP retention during subculture.** To determine the plasmid retention following subculturing, all transconjugant strains were subcultured sequentially in the absence of antibiotic stress to measure generational loss of the GFP plasmid. Parent cultures were revived, confirmed, and washed as described above, and 100 $\mu$L of washed cells were inoculated into 4 mL of antibiotic-free LBS 3% to create a new subculture. This process was repeated daily to produce four sequential subcultures. Subcultures were incubated at 28°C under 100 rpm of shaking agitation overnight ($\sim$18 h) to reach the stationary phase, equating to an average of 5.42 generations elapsed per subculture (see Table S4 for species-specific generation time data). From each subculture (up to passage number 4), 100 $\mu$L of washed cells was serially diluted (10-fold), spread-plated with glass rattler beads (Zymo Rattler plating beads, 4.5 mm) onto antibiotic-free agar plates (species-specific medium preference; see Table 1), and incubated overnight ($\sim$18 h) at 30°C, in duplicate. Isolated colonies were picked randomly from the spread plates ($N = 50$) and patch-plated onto agar plates amended with 300 $\mu$g mL$^{-1}$ kanamycin. This method was employed to diversify subculture passages on both liquid and solid media. Patches that did not grow on the antibiotic-amended media were deemed nonretentive. Plasmid retention was calculated as the percentage of successful patches following each subculture series.

**Evaluation of complex mixtures.** To determine the success of GFP-based differentiation of tagged vibrios from nontagged vibrios using microscopy, mixed communities were created by combining cultures of *V. alginolyticus*, *V. campbellii*, *V. harveyi*, *V. parahaemolyticus*, and *V. vulnificus*. Communities were prepared by combining equal parts of the above-mentioned species to differentiate one GFP-tagged strain

from the other four nontagged species in the mixture. Mixture 1 contained GFP-tagged *V. alginolyticus*, mixture 2 contained GFP-tagged *V. parahaemolyticus*, and mixture 3 contained GFP-tagged *V. harveyi*. To prepare communities, non-GFP cultures were revived from −80°C storage in 3 mL of antibiotic-free 3% LBS, and GFP-tagged strains were revived in 3 mL of LBS 3% amended with 300 $\mu$g mL$^{-1}$ kanamycin. Cultures were grown overnight (~18 h) at 30°C with 100 rpm of shaking agitation. Following growth, 1 mL of each culture was pelleted by centrifugation at ~4,000 × *g*, supernatant fluid was discarded, and pellets were resuspended in 1 mL of sterile 1× PBS, in duplicate, to wash the cells. Next, 100 $\mu$L of each culture was combined in a 1.5-mL centrifuge tube and vortexed for 1 min to homogenize the mixture. Aliquots (5 $\mu$L) of the mixed community were observed using light and fluorescence microscopy to confirm the localization of the GFP strain among the complex mixture (Fig. 1, Fig. S14, and S15).

To evaluate the potential of interspecies plasmid mobilization, a mixture of nontagged *V. vulnificus* and *V. cholerae* and GFP-tagged *V. parahaemolyticus* was combined in coculture to determine if pVSV102 could self-mobilize into nontagged strains. Non-GFP cultures were revived from frozen stocks in 3 mL of antibiotic-free LBS 3%, and GFP-tagged cultures were revived in 3 mL of LBS 3% amended with 300 $\mu$g mL$^{-1}$ kanamycin incubated at 30°C overnight (~18 h). Following incubation, 1 mL of each culture was removed and pelleted by centrifugation at ~4,000 × *g*; the supernatant fluid was discarded, and the cells were resuspended in 1 mL of sterile 1× PBS, in duplicate, to wash the cells. Then, 100 $\mu$L of each culture was combined in a 1.5-mL centrifuge tube and vortexed for 30 s to homogenize the mixture; 100 $\mu$L of the mixture was inoculated into 4 mL of antibiotic-free LBS 3% broth and incubated at 30°C for 5 days. Daily, a 100-$\mu$L aliquot of the mixture was removed, serially diluted (10-fold), and spread-plated with glass rattler beads (Zymo Rattler plating beads, 4.5 mm) onto CHROMagar *Vibrio* plates incubated overnight (~18 h) at 35°C. Following incubation, CFU were counted, speciated colorimetrically, and checked for GFP using a 495-nm black light and fluorescence microscopy. Evidence of interspecies self-mobilization was quantified as the number *V. cholerae* and *V. vulnificus* CFU that acquired GFP in the absence of antibiotic stress.

## SUPPLEMENTAL MATERIAL

Supplemental material is available online only.
**SUPPLEMENTAL FILE 1**, PDF file, 1.1 MB.
**SUPPLEMENTAL FILE 2**, XLSX file, 0.02 MB.

## ACKNOWLEDGMENTS

We are very grateful to Eric Stabb and the Stabb Lab at the University of Illinois Chicago for providing the helper and donor *E. coli* strains and plasmids used to facilitate GFP tagging in this study.

We also thank Samantha Weatherly, Carolina Melendez Declet, and Charlyn Shue for their contributions in the laboratory.

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
