## [Reviewer comments · Microbiology Spectrum]

Microbiology Spectrum

Use and evaluation of a pES213-derived plasmid for the constitutive expression of *gfp* protein in pathogenic vibrios: a tagging tool for *in vitro* studies

William Norfolk and Erin Lipp

Corresponding Author(s): Erin Lipp, University of Georgia

Review Timeline:

Submission Date:	July 18, 2022
Editorial Decision:	September 2, 2022
Revision Received:	October 25, 2022
Accepted:	November 20, 2022

Editor: Minsu Kim

Reviewer(s): The reviewers have opted to remain anonymous.

Transaction Report:

DOI: <https://doi.org/10.1128/spectrum.02490-22>

September 2, 2022

Dr. Erin K Lipp
University of Georgia
Dept. of Environmental Health Science
206 Environmental Health Science Bldg.
Athens, GA 30602-2102

Re: Spectrum02490-22 (Use and evaluation of a pES213-derived plasmid for the constitutive expression of *gfp* protein in pathogenic vibrios: a tagging tool for *in vitro* studies)

Dear Dr. Erin K Lipp:

Thank you for submitting your manuscript to Microbiology Spectrum. Your manuscript was reviewed by two referees. Please see their comments below. While they find that your work would be useful to the community, some important points are raised. We request that you constructively address these concerns in the form of a revised manuscript.

Link Not Available

Sincerely,

Minsu Kim

Journals Department
Reviewer comments:

Reviewer #1 (Comments for the Author):

The authors present data on the conjugation of a KmR plasmid with GFP (pVSV102) into 14 different *Vibrio* strains. An assessment of transfer is shown and the influence of different culture conditions and passages on plasmid stability is presented. The paper is well written and easy to follow. There are, however, some concerns with the work.

The study aims to present an alternative to p15a-based plasmids for long-term tagging of *Vibrio* species that will be maintained in the absence of selection. The authors make a strong point about the many uses for such fluorescently tagged strains in

studying various in situ parameters. The authors go on to state that "The tagging methods described here and the resulting transconjugant strains provide important foundational tools to facilitate experimentation requiring the use of traceable vibrio strains." Also that they "present a simple protocol for GFP tagging of vibrios using a pES213-derived donor plasmid system." While this is an interesting premise, it is worth considering several points as follows:

- 1) The plasmid used in this study was already made and was published on to demonstrate its utility over p15a-based vectors. It seems that this part of the work is expected given the nature of the plasmid.
- 2) The reviewer is not entirely certain what foundational tools are presented in this work that are not already cited in the introduction. Fluorophores have been used to tag Vibrios for in situ experimentation for more nearly three decades in many different systems.
- 3) What simple protocol was presented that is not already highly published on in Vibrio research? The authors use a standard triparental conjugation, which is routinely employed all over the world for Vibrio work (except for the lucky labs that can use natural competence or electroporation).

In short, the authors use an existing plasmid to verify a known premise using an accepted methodology. Other than assessing the curing behavior of this plasmid in a 12 Vibrio strains, this reviewer is unsure what other novel work is presented. It is valuable to know the general curing behavior of a plasmid though the actual curing times were not calculated. A manuscript based on assessing the curing behavior of single plasmid in 12 strains does not make a complete study. The authors could have calculated the actual curing time of the plasmid or demonstrated that curing on ASW was less frequent due to decreased growth rates by comparing doubling times. The manuscript makes a strong point about how useful fluorescently tagged Vibrios would be for in situ studies and then fails to demonstrate the utility of their work in relevant in situ studies. Without additional experimentation, there is not a lot of novelty or reproducibility in this work.

The work presents a simple protocol but does not quantify the efficiency of conjugation or its reproducibility, which is particularly concerning given the presentation of a plate of unsuccessful mating mixtures (Figure 5). The authors do not present how many CFU of each strain are mixed together for successful conjugations. Why were the conjugations incubated for 24-48h? What happened between 24 and 48h that was useful to the efficacy of the conjugation? A small plasmid will conjugate in minutes, so what use is the second 24h interval? If it is simply for the mating spots to start fluorescing, that is not a good measure of conjugal efficiency and is anything but quantitative (nor will it work for other, more subtle reporters). The methodology entirely lacks standardization and reproduction is shown. The reader simply knows that 12 out of 14 Vibrio species received the plasmid, which is inconclusive regarding the ease of the method.

Here are some additional points:

The authors use a pES213-based vector due to its demonstrated lack of curing in *A. fischeri*. This is not the only plasmid used in Vibrio that hangs on without selection - Ushijima et al. (2012, PLoS One, 7(10):e46717; and subsequent papers from that lab) use pRL1383a (an *rsf1010* origin) and its derivatives because they are not found to cure at an appreciable rate. So, there are more options out there for plasmid origins that persist without selection in Vibrios.

Given the emphasis of the manuscript on in situ applications in mixed communities for studies over time, the choice of a pES213-based vector is an interesting one because pES213 was found to mobilize among host cells using an endogenous *A. fischeri* mechanism from pES100. In this case, it is possible to envision the mobilization of a pES213-based vector into other cells in a complex in situ experiment (because the self-mobilization potential of most Vibrio species is unknown), which represents a real and complicated confound. You might see fluorescence where you don't want it.

The use of TBCS as a counter-selection to remove *E. coli* can be dicey. TCBS can greatly decrease the viable CFU of some environmental Vibrio species, so the potential for isolating a positive transconjugant is nonexistent if the conjugal efficiency is already low. Work using auxotrophic *E. coli* in Vibrio conjugations gets around this problem nicely and may increase the likelihood of getting transconjugants (Le Roux, et al. (2007) AEM, 73(3):777-84).

The authors may wish to consider their work on *V. cholerae*. While they postulate on the means of plasmid clearance, work has been published elucidating a mechanism (Jaskólska M, et al. (2022) Nature, 604(7905):323-329). Additionally, labs that work on *V. cholerae* generally introduce constructs into the chromosome, so a plasmid-based system may not catch on.

Reviewer #2 (Comments for the Author):

This manuscript describes the results of experiments to test whether a GFP-encoding plasmid that has been demonstrated to be very stably maintained in *Vibrio fischeri* is also stable in other vibrios. While this plasmid or its derivatives have been used in vibrios other than *V. fischeri*, there has not been a broad study to explore whether these plasmids can be conjugated into, and stably maintained in, a range of vibrios. Fourteen strains were tested.

I believe the information presented in this manuscript will be useful to those who wish to stably mark vibrios with GFP for their studies. I felt the manuscript was well written, although I do have a few comments related to increasing clarity that are listed

below.

1. Line 149. I know in the materials and methods, the procedure for carrying out the subculture experiments is outlined. It was not entirely clear whether the washed cultures were resuspended in an equal amount of PBS. I am assuming yes, but it could be helpful to clarify in the text. Knowing the cultures were stationary and the washed resuspension volume, would it be possible to estimate the number of generations per subculture for the various strains and report this in the manuscript? If possible, this would be useful information for the reader.
2. One bit of information that could be useful to add to the discussion is bringing up that it is known that plasmids like pVSV102 can be transmitted between vibrios (intra and interspecies-Dunn 2005 has some information on this). I liked that the Authors brought up other potential experimental considerations (like adding the plasmid that expresses GFP could result in fitness costs that should be explored prior to experimentation). I think it would be useful to make a short mention about potential transmissibility so that anyone thinking about using this plasmid tool will also have that on their radar.
3. Table 1: I found the column heading "GFP Transfer Concentration" confusing. It seems this refers to the amount of kanamycin used as a selection for vibrios that acquired the plasmid via conjugation? If this is correct, I would suggest changing the wording to reflect this better. Also, are the amounts listed in the MIC and GFP transfer columns possibly switched? It was not clear to me why one would select for cells containing a plasmid using less than the MIC.
4. Figure 3: Should the sentence on line 581 be "fluorescent" rather than "non-fluorescent"? On line 583, it seems it could be too strong of wording to indicate that the two strains did not survive. Instead, maybe they were not recoverable as CFUs?
5. Figure 4: I was a little confused by this figure. Why were these patched instead of just looking at CFUs? And why was 300 ug/ml kanamycin used (thinking about the kanamycin levels presented in Table 1)? The y-axis label indicates this is fluorescent CFU data. I think having more of an explanation for why this experiment differed from the others would be helpful.
6. Figure 5: I was not entirely sure of the significance of this figure. I would expect that the donor E. coli with pVSV102 would be fluorescent, so one might expect that there would be some signal from conjugation spots. How can one be sure that the fluorescence is from a vibrio? And why did the spots have GFP signal in one conjugation attempt but not the other plate? I feel that this figure could be deleted unless there is more explanation added in the text to clarify its significance. I will say that the spots and the GFP distribution are cool, though.

Staff Comments:

Preparing Revision Guidelines

Please return the manuscript within 60 days; if you cannot complete the modification within this time period, please contact me. If you do not wish to modify the manuscript and prefer to submit it to another journal, please notify me of your decision immediately so that the manuscript may be formally withdrawn from consideration by Microbiology Spectrum.

Corresponding authors may join or renew ASM membership to obtain discounts on publication fees. Need to upgrade your

membership level? Please contact Customer Service at Service@asmusa.org.

Response to Reviewers
(authors' responses are in red)

Reviewer #1 (Comments for the Author):

The authors present data on the conjugation of a KmR plasmid with GFP (pVSV102) into 14 different *Vibrio* strains. An assessment of transfer is shown and the influence of different culture conditions and passages on plasmid stability is presented. The paper is well written and easy to follow. There are, however, some concerns with the work.

The study aims to present an alternative to p15a-based plasmids for long-term tagging of *Vibrio* species that will be maintained in the absence of selection. The authors make a strong point about the many uses for such fluorescently tagged strains in studying various in situ parameters. The authors go on to state that "The tagging methods described here and the resulting transconjugant strains provide important foundational tools to facilitate experimentation requiring the use of traceable *vibrio* strains." Also that they "present a simple protocol for GFP tagging of *vibrios* using a pES213-derived donor plasmid system." While this is an interesting premise, it is worth considering several points as follows:

1) The plasmid used in this study was already made and was published on to demonstrate its utility over p15a-based vectors. It seems that this part of the work is expected given the nature of the plasmid.

Thank you for your comment. While prior publications have employed this plasmid, no formal comparison of its efficacy or curing in multiple *vibrios* has been published and thus is the target of our research.

2) The reviewer is not entirely certain what foundational tools are presented in this work that are not already cited in the introduction. Fluorophores have been used to tag *Vibrios* for in situ experimentation for more nearly three decades in many different systems.

We have revised the phrasing of "foundational tools" to be "standardized tools" to better indicate that these methods are a standardization rather than a novel method. See revised manuscript lines 47-49: "These methods and the resulting transconjugant strains provide important standardized tools to facilitate experimentation requiring the use of traceable *vibrio* strains."

3) What simple protocol was presented that is not already highly published on in *Vibrio* research? The authors use a standard triparental conjugation, which is routinely employed all over the world for *Vibrio* work (except for the lucky labs that can use natural competence or electroporation).

As the reviewer indicates, tri-parental mating is a well-established method; however, the protocol presented here describes a standardized culture-based workflow (including media types, incubation times, inoculum concentrations) for this assay to be readily adaptable for multiple species of *Vibrio*. Our intention was not to highlight the use of tri-parental conjugation but rather to compile the conditions for which this assay can be effectively applied to a range of

vibrios, using the same basic structured protocol and predictable laboratory consumables. Furthermore, the determination of species-specific antibiotic tolerance concentrations is a laborious process for which these methods designate an optimal value for each *Vibrio* spp. tagged through this work.

This intention is clarified in the importance section of the revised manuscript at lines 45-49: “Here we present a structured protocol for conjugation-based tagging of vibrios using the pES213-derived plasmid pVSV102 and evaluate the plasmid stability of tagged strains. These methods and the resulting transconjugant strains provide important standardized tools to facilitate experimentation requiring the use of traceable vibrio strains.”

In short, the authors use an existing plasmid to verify a known premise using an accepted methodology. Other than assessing the curing behavior of this plasmid in a 12 *Vibrio* strains, this reviewer is unsure what other novel work is presented. It is valuable to know the general curing behavior of a plasmid though the actual curing times were not calculated. A manuscript based on assessing the curing behavior of single plasmid in 12 strains does not make a complete study. The authors could have calculated the actual curing time of the plasmid or demonstrated that curing on ASW was less frequent due to decreased growth rates by comparing doubling times. The manuscript makes a strong point about how useful fluorescently tagged *Vibrios* would be for *in situ* studies and then fails to demonstrate the utility of their work in relevant *in situ* studies. Without additional experimentation, there is not a lot of novelty or reproducibility in this work.

Part of the goal for this work was to assess the efficacy and rate of plasmid loss for each of the tagged *Vibrio* strains in order to determine their utility for *in situ* research. While an assessment of the full plasmid curing is useful information, a loss of $\geq 20\%$ of total plasmid retention would add substantial bias to any *in situ* study. Therefore, we focused on determining the maximum length of time that each transconjugant strain can be expected to retain the GFP plasmid in the absence of antibiotic stress rather than assessment of full curing. Furthermore, due to the observed stability of pVSV102, we hypothesize that the full curing of this plasmid may take several weeks to months depending on the *Vibrio* spp. and conditions of culture maintenance.

Additionally, three of the resulting GFP strains (*V. alginolyticus*, *V. harveyi*, and *V. mediterranei*) produced through this work have been successfully applied to investigate the mechanisms of coral disease transmission using *in situ* experimentation. This research is focused on the ecological mechanisms that contribute to *Vibrio* colonization from ambient seawater rather than the specific GFP tagging protocol and thus is being prepared as a separate publication that will cite the present research.

The work presents a simple protocol but does not quantify the efficiency of conjugation or its reproducibility, which is particularly concerning given the presentation of a plate of unsuccessful mating mixtures (Figure 5).

We appreciate that this figure caused some confusion. This unsuccessful mating mixture image (Figure 5 in original submission) was provided as an example to show a distinction between a potentially successful mixture and an unsuccessful one. Unsuccessful transfer of the plasmid

occurred in this image due to the use of a kanamycin concentration that was intolerable for the target *Vibrio* (concentration was too high, which likely inhibited growth of the target *Vibrio*). We clarified this in the caption and in the methods section (revised version lines 327 and 334-335) that these visible fluorescent patches are “indicative of successful transfer” of the GFP plasmid, but confirmation must be accomplished with microscopy following purification of the tagged strain on TCBS agar. To avoid additional confusion, this figure has been moved from the main text to supplemental documentation (revised version Figure S1).

To address the question of efficacy, in the revised manuscript we included data showing the rate of conjugation efficiency (N = 24 mixtures) for all successfully tagged vibrios in this study. The results of this work demonstrated strong transfer efficacy (≥ 10 mixtures successful for all target vibrios) when mated triparentally on kanamycin amended media equal to the stress concentration listed in Table 1. We have also added text regarding this to the methods and results sections at lines 336-337 and 127-129, respectively, in the revised manuscript. Conjugation success rates for each species has been added to the supplemental material in Table S1 (revised version).

The authors do not present how many CFU of each strain are mixed together for successful conjugations.

Thank you for this suggestion. We have added the mating mixture CFU values as a column in Table 1 and Table 2 in the revised manuscript.

Why were the conjugations incubated for 24-48h? What happened between 24 and 48h that was useful to the efficacy of the conjugation? A small plasmid will conjugate in minutes, so what use is the second 24h interval?

We found that the extended incubation time for the mating mixtures was needed to offset the decreased growth rate of the target vibrio under the level of kanamycin stress. This has been added to the methods section at lines 324-326: “Extended duration incubation was utilized to account for the reduced rate of growth of the target vibrio under the given level of kanamycin stress.”

If it is simply for the mating spots to start fluorescing, that is not a good measure of conjugal efficiency and is anything but quantitative (nor will it work for other, more subtle reporters). We agree with the reviewer’s observation and have attempted to better qualify the use of the mating spots as an initial check. As stated above, we have edited the wording of the methods section (lines 327 and 334-335) to clarify that fluorescence of the mating spots was used to indicate conjugation success, but confirmation must be accomplished using microscopy. Furthermore, we have added a note to the caption of Figure S1 (revised version) that other more subtle reporters may not be amenable to this method of detection.

The methodology entirely lacks standardization and reproduction is shown. The reader simply knows that 12 out of 14 *Vibrio* species received the plasmid, which is inconclusive regarding the ease of the method.

We believe that this work presents an easy-to-follow workflow that standardizes the culture-based methods required to produce GFP-tagged *Vibrio* strains using the pVSV102 plasmid in a standard microbiology laboratory. We have attempted to address the reviewer's specific concerns (as noted above) that should improve information needed for others to reproduce this protocol.

Here are some additional points:

The authors use a pES213-based vector due to its demonstrated lack of curing in *A. fischeri*. This is not the only plasmid used in *Vibrio* that hangs on without selection - Ushijima et al. (2012, PLoS One, 7(10):e46717; and subsequent papers from that lab) use pRL1383a (an rsf1010 origin) and its derivatives because they are not found to cure at an appreciable rate. So, there are more options out there for plasmid origins that persist without selection in *Vibrios*.

We have added text discussing other plasmid alternatives to the discussion section citing these works. See revised manuscript at lines 184-188: "It should be noted, that while this research prioritizes the use of pVSV102 in contrast to p15A-derivatives, other plasmid alternatives such as pRL1383a (Ushijima et al., 2012; Ushijima et al., 2014), pUTat (Xaio et al., 2011), pET28a (Dai et al., 2020), and pRK600 (Pollock et al., 2015) have also been shown to persist stably in *vibrios*."

Given the emphasis of the manuscript on in situ applications in mixed communities for studies over time, the choice of a pES213-based vector is an interesting one because pES213 was found to mobilize among host cells using an endogenous *A. fischeri* mechanism from pES100. In this case, it is possible to envision the mobilization of a pES213-based vector into other cells in a complex in situ experiment (because the self-mobilization potential of most *Vibrio* species is unknown), which represents a real and complicated confound. You might see fluorescence where you don't want it.

We appreciate the reviewer calling this out. To address this potential confounder, we conducted an additional experiment to assess the ability of pVSV102 to mobilize without antibiotic stress into other non-tagged *Vibrio* strains. Briefly, this experiment combined non-tagged *V. cholerae* and *V. vulnificus* with GFP-tagged *V. parahaemolyticus* in a co-culture for a period of 5 days. Daily, an aliquot of the culture was removed, serially diluted, and spread plated onto CHROMAgar *Vibrio* (for rapid species identification/quantification). The number of fluorescent colonies of each species was counted and used to determine a frequency of interspecies mobilization.

The results of this experiment showed that pVSV102 did not mobilize into *V. cholerae* or *V. vulnificus* from GFP-tagged *V. parahaemolyticus* in an antibiotic free complex mixture. Text associated with this experiment was added to the methods, results, and discussion at lines 399-414, 133-134, and 189-198, respectively.

The use of TBCS as a counter-selection to remove *E. coli* can be dicey. TCBS can greatly decrease the viable CFU of some environmental *Vibrio* species, so the potential for isolating a positive transconjugant is nonexistent if the conjugal efficiency is already low. Work using

auxotrophic *E. coli* in *Vibrio* conjugations gets around this problem nicely and may increase the likelihood of getting transconjugants (Le Roux, et al. (2007) AEM, 73(3):777-84).

As the reviewer indicates, TCBS can be a challenging media for some vibrio. However, the use of TCBS to selectively remove *E. coli* has been successfully applied in prior *Vibrio*-GFP tagging studies (e.g., O'Toole et al., 1996 Molecular Microbiology 19(3), 625–637; O'Toole et al., 2004 Microbial Pathogenesis 37(1): 41-46; Durai et al., 2011 Journal of Basic Microbiology 51: 243-252). We have added additional clarifying text to the methods section to describe that while TCBS isolation is sufficient to counter select *E. coli*, the growth of some vibrios may be suboptimal on this media. See revised manuscript lines 340-342: "It should be noted that while TCBS agar is valid for the removal of *E. coli*, this media does not always produce optimal growth for some vibrio species, thus working stocks of these cultures should be maintained on LBS amended with 300 $\mu\text{g mL}^{-1}$ kanamycin once isolated."

Additionally, we have also added a note about the use of auxotrophic *E. coli* as an alternative to TCBS selection at lines 342-344: "For vibrios that are not amenable to growth on TCBS, prior research has successfully utilized auxotrophic *E. coli* strains to enable selective removal following conjugation (Le Roux et al., 2007)."

The authors may wish to consider their work on *V. cholerae*. While they postulate on the means of plasmid clearance, work has been published elucidating a mechanism (Jaskólska M, et al. (2022) Nature, 604(7905):323-329). Additionally, labs that work on *V. cholerae* generally introduce constructs into the chromosome, so a plasmid-based system may not catch on. The work of Jaskólska et al., (2022) is cited in our discussion (lines 236-238) of the potential issues associated with the tagging of *V. cholerae*.

Reviewer #2 (Comments for the Author):

This manuscript describes the results of experiments to test whether a GFP-encoding plasmid that has been demonstrated to be very stably maintained in *Vibrio fischeri* is also stable in other vibrios. While this plasmid or its derivatives have been used in vibrios other than *V. fischeri*, there has not been a broad study to explore whether these plasmids can be conjugated into, and stably maintained in, a range of vibrios. Fourteen strains were tested.

I believe the information presented in this manuscript will be useful to those who wish to stably mark vibrios with GFP for their studies. I felt the manuscript was well written, although I do have a few comments related to increasing clarity that are listed below.

1. Line 149. I know in the materials and methods, the procedure for carrying out the subculture experiments is outlined. It was not entirely clear whether the washed cultures were resuspended in an equal amount of PBS. I am assuming yes, but it could be helpful to clarify in the text. Knowing the cultures were stationary and the washed resuspension volume, would it be possible to estimate the number of generations per subculture for the various strains and report this in the manuscript? If possible, this would be useful information for the reader.

We have clarified the volume of culture and PBS used to wash the cells in the methods section at lines 311-315: “Following incubation, 1 mL of vibrio and *E. coli* cultures were pelleted by centrifugation at ~4,000 x *g* for 2 min then resuspended in 1 mL of sterile 1X phosphate-buffered saline (PBS). This procedure was repeated twice to wash cells and remove residual media. 100 μ L (colony forming units [CFU] reported in Table 1 and 2) of the washed helper, donor, and target recipient was removed and combined in a 1.5 mL microcentrifuge tube.”

Additionally, we have added an estimate of the average number of generations per subculture to the methods section at lines 372-374: “Subcultures were incubated at 28 °C under 100 rpm of shaking agitation overnight (~18 h) to reach stationary phase equating to an average of 5.42 generations elapsed per subculture (see Table S4 for species-specific generation time data).” A species-specific breakdown of the generations estimate has also been added to the supplemental documentation as Table S4.

2. One bit of information that could be useful to add to the discussion is bringing up that it is known that plasmids like pVSV102 can be transmitted between vibrios (intra and interspecies- Dunn 2005 has some information on this). I liked that the Authors brought up other potential experimental considerations (like adding the plasmid that expresses GFP could result in fitness costs that should be explored prior to experimentation). I think it would be useful to make a short mention about potential transmissibility so that anyone thinking about using this plasmid tool will also have that on their radar.

To address this potential confounder, we have added an additional experiment which assessed the ability of pVSV102 to mobilize without antibiotic stress into other non-tagged *Vibrio* strains (described above). Text describing the details of this experiment results have been added to the methods, results, and discussion at lines 399-414, 133-134, and 189-198, respectively.

3. Table 1: I found the column heading "GFP Transfer Concentration" confusing. It seems this refers to the amount of kanamycin used as a selection for vibrios that acquired the plasmid via conjugation? If this is correct, I would suggest changing the wording to reflect this better. Also, are the amounts listed in the MIC and GFP transfer columns possibly switched? It was not clear to me why one would select for cells containing a plasmid using less than the MIC.

To avoid confusion, we have changed the “GFP Transfer Concentration” column to read as “Stress Concentration” to match the wording of the methods section. The stress concentration is the kanamycin concentration that induces a “stressful but non-lethal” environment for the growth of the target *Vibrio*. This concentration was used to facilitate transfer of the GFP plasmid when combined with the helper and donor *E. coli* strains in the mating mixture. We have also added a sub-caption to Table 1 to clarify the difference between the MIC and stress concentration when viewing the table.

4. Figure 3: Should the sentence on line 581 be "fluorescent" rather than "non-fluorescent"? On line 583, it seems it could be too strong of wording to indicate that the two strains did not survive. Instead, maybe they were not recoverable as CFUs?

Thank you for pointing out this discrepancy. We have corrected the wording error in the caption of Figure 3 and revised our commentary of *V. pelagius* and *V. splendidus* to be “not

recovered” instead of “did not survive.” This revision has also been corrected in the results section at lines 153-154.

5. Figure 4: I was a little confused by this figure. Why were these patched instead of just looking at CFUs? And why was 300 ug/ml kanamycin used (thinking about the kanamycin levels presented in Table 1)? The y-axis label indicates this is fluorescent CFU data. I think having more of an explanation for why this experiment differed from the others would be helpful.

We acknowledge that this figure caused confusion as presented. Patch plating was employed in this evaluation to expose the culture to both a liquid and solid media for each passage step. This information has been added to the methods section at lines 379-380: “This method was employed to diversify subculture passages on both liquid and solid media.”

The final kanamycin concentration of 300µg/mL was used throughout this study to maintain GFP-tagged Vibrio strains. This concentration is well above the maximum MIC for all non-tagged vibrio species tested and thus would select against any non-retentive patches applied to the plate.

6. Figure 5: I was not entirely sure of the significance of this figure. I would expect that the donor E. coli with pVSV102 would be fluorescent, so one might expect that there would be some signal from conjugation spots. How can one be sure that the fluorescence is from a vibrio? And why did the spots have GFP signal in one conjugation attempt but not the other plate? I feel that this figure could be deleted unless there is more explanation added in the text to clarify its significance. I will say that the spots and the GFP distribution are cool, though.

We acknowledge that this figure caused confusion as presented. The significance of this figure was to show the appearance of a potentially successful mating mixture in contrast to an unsuccessful one to provide the reader with a simple lab-based check that could be used to determine if the mating mixture was ready to progress to purification on TCBS. To clarify the interpretation of this figure, we have added additional context to the figure caption explaining the reason for the unsuccessful conjugation attempt (improper antibiotic stress concentration). We have also added text to the methods section at lines 327 and 334-335 clarifying the interpretation of the fluorescent patches as “potentially successful” conjugations. Additionally, this figure has been moved from the main text to supplemental documentation and is now designated Figure S1.

November 20, 2022

Dr. Erin K Lipp
University of Georgia
Dept. of Environmental Health Science
206 Environmental Health Science Bldg.
Athens, GA 30602-2102

Re: Spectrum02490-22R1 (Use and evaluation of a pES213-derived plasmid for the constitutive expression of *gfp* protein in pathogenic vibrios: a tagging tool for *in vitro* studies)

Dear Dr. Erin K Lipp:

Congratulation! Your manuscript has been accepted, and I am forwarding it to the ASM Journals Department for publication. You will be notified when your proofs are ready to be viewed.

Sincerely,

Minsu Kim
Editor, Microbiology Spectrum

Journals Department
Supplemental Material: Accept
Supplemental Dataset: Accept